# Evidence for spin-dependent energy transport in a superconductor

M. Kuzmanović [1], B. Y. Wu[1,2], M. Weideneder[1,3], C. H. L. Quay [1✉] & M. Aprili[1]

In ferromagnetic materials, spin up and down electrons can carry different heat currents. This spin-dependent energy excitation mode ('spin energy mode') occurs only when spin up and down energy distribution functions are different. In superconductors, heat is carried by quasiparticle excitations and the spin energy mode can be excited by spin-polarised current injection. In the presence of a finite Zeeman magnetic field, the spin energy mode surprisingly leads to a charge imbalance (different numbers of hole- and electron-like quasiparticles) at the superconducting gap edge. By performing spin-resolved spectroscopy of the out-of-equilibrium quasiparticle populations in a mescoscopic superconductor, we reveal that their distribution functions are non-Fermi–Dirac. In addition, our spectroscopic technique allows us to observe a charge imbalance, localised in energy to the gap edge and thus unambiguously identify the spin energy mode. Our results agree well with theory and shed light on energy transport in superconducting spintronics.

[1] Laboratoire de Physique des Solides (CNRS UMR 8502), Université Paris-Saclay, 91405 Orsay, France. [2] Graduate Institute of Applied Physics, National Taiwan University, Taipei 10617, Taiwan. [3] Institute for Experimental and Applied Physics, University of Regensburg, 93053 Regensburg, Germany. ✉email: charis.quay@universite-paris-saclay.fr

The Seebeck effect, in which a temperature gradient leads to a charge current, was first observed about two centuries ago. Together with its Onsager reciprocal, the Peltier effect, it forms the basis of the field of thermoelectricity or coupled charge and heat transport[1]. Coupled charge and spin transport, or spintronics, emerged in the late 1980s[2]. Later, spin caloritronics or coupled heat, charge and spin transport[3,4] became an experimental reality with the observation of the spin Seebeck effect[5] and spin-dependent Peltier effects[6] in normal metals, and very recently large spin-dependent thermoelectric effects in superconductor-based devices[7–11].

Early work in spin caloritronics focused on temperature differences across interfaces between (magnetic) materials, shown to be associated with spin and/or charge currents. Within a given material, it was pointed out that spin up and down carriers (electrons or quasiparticles (QPs)) can also have different temperatures[12–18]. When this happens, the spin energy mode of the system is excited and the two spin species carry different heat currents. Evidence for spin-dependent heat transport was recently observed in a normal metal[19], but not in superconductors. Moreover, due to the aggregate nature of the measurements in normal metals (giant magnetoresistance of a spin valve), detailed information on the distribution function could not be obtained.

Here, we study thin-film superconducting aluminium. As our measurements are spectroscopic, we are able to reveal QP populations which cannot be described by effective temperatures. Instead, they carry an 'imprint' of the electron distribution function in the normal metal from which current is injected into the superconductor, to generate QPs. Further, unlike in normal metals, the spin energy mode in superconductors gives rise to a charge imbalance (i.e. different numbers of electron-like and hole-like QPs) with a specific energy and magnetic field dependence. Our spectroscopic measurements allow us to observe this imbalance and thus unambiguously identify the spin energy mode. The presence of the spin energy mode in turn necessarily implies that the distribution functions of spin up and down QPs are different.

## Results

### Out-of-equilibrium superconductors' spinful excitation modes.
The ground state of conventional (Bardeen–Cooper–Schrieffer) superconductors is composed of Cooper pairs of electrons in a spin singlet configuration. In equilibrium, this macroscopic quantum state can carry a dissipationless charge current (known as a supercurrent), but not spin or energy currents. In contrast, the single particle excitations, or QPs, are spin-1/2 fermions, which can carry spin, energy and charge currents. The density of states of these QPs ($\rho(E)$) is zero in an energy range $\pm\Delta$ about the Fermi energy ($E_F$), and has coherence peaks just above this gap (Fig. 1a).

Out-of-equilibrium QP populations in superconductors can be described by the particle energy distribution function $f(E)$. Neglecting the QP spin, $f(E)$ can be decomposed based on symmetry into energy $f_L(E) = f(-E) - f(E)$ and charge $f_T(E) = 1 - f(E) - f(-E)$ modes[20,21]. The simplest $f(E)$ which excites these modes are, respectively, an effective temperature (different from the lattice temperature) and a charge imbalance. In the presence of a charge imbalance, the number of electron-like and hole-like QPs are non-identical, and the QP chemical potential is different from the Fermi energy. Extensive experimental and theoretical work has been done on both charge and energy modes (see ref. [22] and Chapter 11 of ref. [23]).

In the spinful case, the decomposition above can be generalised by the addition of spin and spin energy modes, $f_{T3}(E) = [f_{T\uparrow}(E) - f_{T\downarrow}(E)]/2$ and $f_{L3}(E) = [f_{T\uparrow}(E) - f_{L\downarrow}(E)]/2$[15,16,18]. $f_{L3}$ is most

simply excited by a spin-dependent temperature and $f_{T3}$ by a spin-dependent chemical potential. The spin and spin energy modes only exist if spin up and down QPs have different distribution functions, i.e. if $f_\uparrow(E) \neq f_\downarrow(E)$. By construction, $f_L$ and $f_{L3}$ are odd in energy, while $f_T$ and $f_{T3}$ are even in energy. In the following, we focus mainly on $f_{L3}$, the spin energy mode.

To generate different spin up and down distribution functions, it is necessary to preferentially excite QPs of one spin species. In thin superconducting films, this can be done by applying an in-plane magnetic field ($H$), which lowers (raises) the energy of spin down (up) QPs by the Zeeman energy ($E_Z$) and splits the DOS so that only spin down excitations (spin down electron-like and spin up hole-like QPs) are allowed in the energy range $\Delta - E_Z \leq |E| \leq \Delta + E_Z$ (Fig. 1b) ($E_Z = \mu_B H$, with $\mu_B$ the Bohr magneton). Current injection in this energy range thus creates spin-polarised QPs regardless of the magnetic properties of the tunnel barrier or the injector electrode.

For our experiments, we use thin-film superconducting (S) aluminium wires, with a native insulating (I) oxide layer, across which lie normal metal (N) and superconducting (S') electrodes. The former is used as an injector and the latter as detectors (Fig. 1f). S is terminated on both sides by reservoirs at a distance of about 5 μm from the NIS junction. The magnetic field ($H$) is applied in the plane, parallel to S.

Our basic spectroscopy measurement consists of injecting a constant current ($I_{inj}$) at the injector ($J_{inj}$), and measuring the current ($I_{det}$) and/or the differential conductance ($G_{det} = dI_{det}/dV_{det}$) as a function of the applied voltage ($V_{det}$) at one of the detectors ($J_{det1}$, $J_{det2}$ and $J_{det3}$ in Fig. 1f). Measurements were performed in a dilution refrigerator with a base temperature of 90 mK. $J_{det1}$ lies within both an electron–electron interaction length ($\lambda_{e-e} \approx 1$ μm[24,25]) and a spin–flip length ($\lambda_{sf} \approx 300$ nm[26,27]) of the injector.

We model our system using the Keldysh–Usadel equations, which describe out-of-equilibrium diffusive superconductors (see Supplementary Methods 1.1.1 for details). Following refs. [16,18,28], we solve these numerically in one dimension, assuming negligible (inelastic) electron–electron and electron–phonon interactions, and include a Zeeman magnetic field. Experimental parameters are used in the model: the normal state diffusion constant $D \approx 10$ cm$^2$ s$^{-1}$, $L = 10$ μm, $R(J_{inj}) = 13$ kΩ. The diffusion time from the injector to the reservoirs is $\tau_{diff} = l_{inj-res}^2/D \approx 20$ ns, where $l_{inj-res}$ is the injector–reservoir distance $\approx L/2$. As $\tau_{diff}$ is much smaller than the QP recombination time ($\tau_{rec} \gtrsim 1$ μs[29]), QPs relax and recombine at the reservoirs. At the interface with the injector, the boundary conditions are given by spectral current continuity and the injector distribution function $f_{inj}(E - eV_{inj})$, assumed to be Fermi–Dirac.

In our numerical results for the closest detector (Fig. 1c), we see that the QP distribution function bears signatures of both the density of states in S (Fig. 1b) as well as the distribution function in the injector: It has a peak at $E = \Delta$ and goes sharply to zero at $E = eV_{inj}$, with $e$ the electron charge. The distribution function is also spin-dependent.

To interpret our experimental results, it is also helpful to understand the link between the spin energy mode $f_{L3}$ and charge imbalance by considering the non-equilibrium QP number as a function of energy:

$$
\begin{aligned}
N(E) &= N_\uparrow(E) + N_\downarrow(E) \\
&= \frac{1}{2}[f_\uparrow(E) - f_0(E)]\rho_\uparrow(E) + \frac{1}{2}[f_\downarrow(E) - f_0(E)]\rho_\downarrow(E)
\end{aligned}
\tag{1}
$$

$$
= -\frac{1}{2}[\rho_- f_{L3} + \rho_+ f_T] + \frac{1}{2}[\rho_+(f_L^0 - f_L) - \rho_- f_{T3}]
\tag{2}
$$

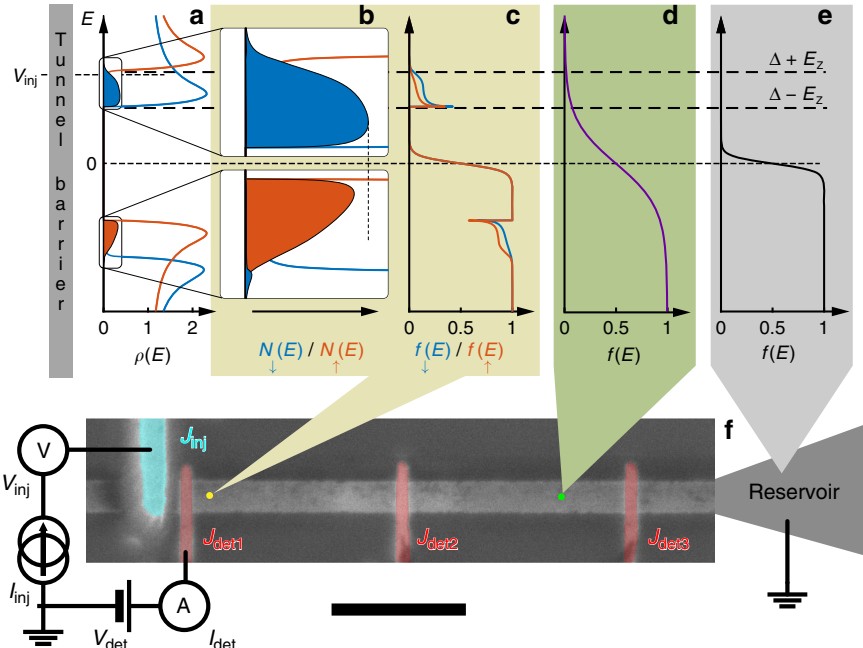

**Fig. 1 Generation and detection of out-of-equilibrium quasiparticles (QP) in a superconductor. a** Spin down (blue) and up (red) QP density of states (DOS) in the superconductor in an in-plane magnetic field, which induces both a Zeeman splitting and orbital depairing. The blue and red shaded regions are proportional to, respectively, the number of spin down and spin up non-equilibrium quasiparticles (as in Eq. (1)) near the first detector. (A spin down excitation is a spin down electron-like or a spin up hole-like quasiparticle.) This was calculated with the density of states in **a**, the reservoir distribution function in **e** and the injection voltage ($V_{\mathrm{inj}}$) indicated on the left. For clarity, the imbalance between the number of electron-like QPs and the number of hole-like QPs (the charge imbalance, i.e. the odd component of $N(E)$), has been multiplied by five. This imbalance can be seen to occur in a specific energy range, $\Delta - E_Z \leq |E| \leq \Delta + E_Z$. **b** Zoom in of **a**. It can be seen more clearly here that there are more quasiparticles of one spin (blue) than the other (red). **c** Predicted spin down (blue) and up (red) QP distribution functions at the indicated distance from the injector. The distribution functions show peaks at the superconducting gap edge, as well as a step-like cutoff at $eV_{\mathrm{inj}}$. **d** Farther than an electron–electron interaction length ($\approx$1 µm) from the injector, we expect the quasiparticle distribution function to be spin-independent and close to an effective temperature. The trace shown here is an illustration, not a calculation. **e** QPs are assumed to be at equilibrium at the reservoir. **f** False colour scanning electron micrograph of the device, and a schematic drawing of the spectroscopy measurement setup. The horizontal superconducting wire is 6 nm Al. The injector (100 nm Cu, cyan) and the detectors (8 nm Al/0.1 nm Pt, red) form tunnel junctions with the wire, with the latter's native oxide as the barrier. Scale bar: 1 µm.

Here $\rho_\uparrow(E)$ and $\rho_\downarrow(E)$ are the DOS of spin up and spin down QPs, respectively; $\rho_+(E) \equiv \frac{1}{2}[\rho_\uparrow(E) + \rho_\downarrow(E)] = \rho(E)$; $\rho_-(E) \equiv \frac{1}{2}[\rho_\uparrow(E) - \rho_\downarrow(E)]$; and $f_0(E)$ and $f_L^0(E)$ are, respectively, $f(E)$ and $f_L(E)$ at equilibrium.

In Eq. (2) we notice that the term $\rho_-(E)f_{L3}(E)$ is even in energy, which means that the spin energy mode $f_{L3}$ adds particles at both positive and negative energies, and raises the overall QP chemical potential, thus creating a charge imbalance. (The first (last) term in Eq. (2) is even (odd) in energy and creates a charge (energy) imbalance.) (Fig. 1b) In addition, the factor $\rho_-(E)$ means that $f_{L3}$ add particles in the energy range $\Delta - E_Z \leq |E| \leq \Delta + E_Z$, regardless of the injection voltage or other experimental parameters (Fig. 1a and b). $f_T$ also creates a charge imbalance, which however appears at low magnetic fields and high energies (Supplementary Fig. 6 and Supplementary Discussion 2.5). Our spectroscopic technique allows us distinguish between $f_{L3}$ and $f_T$, based on their different energy dependences. We refer the reader to refs. [16,18] and Supplementary Methods 1.2 for further theoretical details.

**Spectroscopic spin-sensitive QP detection.** We first characterise both injector and detector junctions, and explain our spectroscopy technique. Figure 2a shows the differential conductance of the injector ($G_{\mathrm{inj}} = \mathrm{d}I_{\mathrm{inj}}/\mathrm{d}V_{\mathrm{inj}}$) as a function of the applied voltage ($V_{\mathrm{inj}}$) at different magnetic fields ($H$). At the temperatures of our experiment, $G_{\mathrm{inj}}$ is almost exactly proportional to the density of states in S[23]. We can see that $H$ induces Zeeman splitting of the QP density of states. $H$ also couples to the orbital degree of

freedom, inducing screening supercurrents and hence a rounding of the QP coherence peak due to orbital depairing[23,30]. From fits to the data, we obtain an Abrikosov–Gor'kov orbital depairing parameter of $\alpha = R_{\mathrm{ORB}}H^2$, with $R_{\mathrm{ORB}} \approx 6.5$ µeV T$^{-2}$. The critical field of S is $\approx$2.7 T. In the results shown here, the Zeeman energy is always greater than the depairing parameter (see Supplementary Section 2.1 for details).

If the detector temperature is much smaller than the super-conducting energy gap in S' ($k_B T_{\mathrm{det}} \ll \Delta_{\mathrm{det}}$, with $k_B$ Boltzmann's constant), the differential conductance of SIS' junctions as a function of the applied voltage in the subgap region ($V_{\mathrm{det}} < (\Delta + \Delta_{\mathrm{det}})/e$) is given by

$$G_{\mathrm{det}}(V_{\mathrm{det}}) = \frac{1}{eR_N} \int N(E) \frac{\partial \rho_{\mathrm{det}}(E + eV_{\mathrm{det}})}{\partial V_{\mathrm{det}}} \mathrm{d}E \qquad (3)$$

where $\rho_{\mathrm{det}}(E)$ the density of states in S', $N(E)$ the QP number from Eqs. (1) and (2), and $R_N$ the normal state resistance of the detector junction.

Most of the integral in Eq. (3) comes from the coherence peak in $\rho_{\mathrm{det}}(E)$ at $E = \Delta_{\mathrm{det}}$. This peak picks out $N(E)$ the number of QPs in S, shifted by $\Delta_{\mathrm{det}}$. In other words, $G_{\mathrm{det}}(V_{\mathrm{det}} - \Delta_{\mathrm{det}}/e)$ gives the number of QPs at energy $E = eV_{\mathrm{det}}$, while $I_{\mathrm{det}}(V_{\mathrm{det}} - \Delta_{\mathrm{det}}/e)$ gives the total number of QPs for $E \leq eV_{\mathrm{det}}$. Our measurements thus give us spectroscopic information on the QPs (see Supplementary Methods 1.2.2 for details).

Charge imbalances in the QP population lead to an even-in-energy $N(E)$. As $\frac{\partial \rho_{\mathrm{det}}(E + eV_{\mathrm{det}})}{\partial V_{\mathrm{det}}}$ is odd in $E$, and the convolution of an

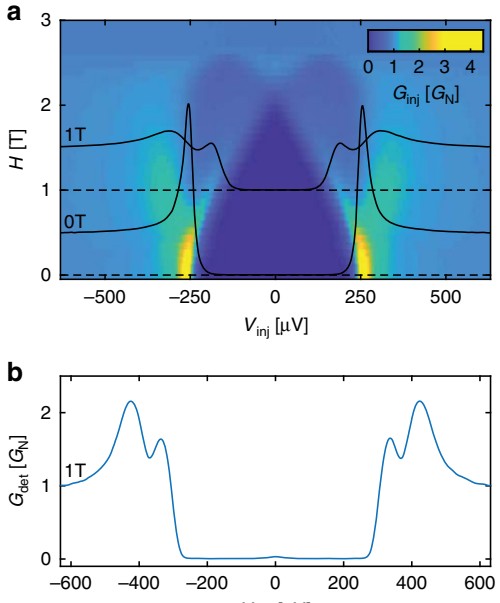

**Fig. 2 Characterisation of injector and detector junctions. a** Differential conductance of the injector junction ($G_{inj}$) as a function of injector voltage ($V_{inj}$) and magnetic field ($H$), and horizontal slices at $H = 0$ T and $H = 1$ T (black traces). The quasiparticle density of states in the superconducting wire can be seen to be clearly Zeeman split at 1T. **b** Differential conductance of the detector junction ($G_{det}$) as a function of the detector voltage ($V_{det}$) at $H = 1$ T without any injection current. As the detector is not Zeeman-split, we also see the Zeeman splitting of the quasiparticle density of states in the superconducting wire.

even and an odd function gives an odd function, we expect a charge imbalance to lead to a component of $G_{det}(V_{det})$ which is odd in $V_{det}$.

At finite magnetic fields, these spectroscopic measurements become spin-sensitive if Zeeman spin-splitting occurs in S but not in S′; the unsplit coherence peak in S′ separately probes the number of excitations in S at the two gap edges for spins up and down, respectively, at $V_{det}^{\uparrow(\downarrow)} = |\Delta \pm E_Z - \Delta_{det}|/e$.

We suppress the spin-splitting in S′ through the strong spin–orbit coupling of sprinkled Pt, which acts as a spin-mixer (see "Methods" section, Supplementary Methods 1.2.3 and refs. [30–33]). Figure 2b shows $G_{det}(V_{det})$ at different $H$ and $I_{inj} = 0$. At $H = 1$ T, we see two peaks, as expected for a non-spin-split detector. (Were there a Zeeman splitting in S′ equal to that in S, the situation would be equivalent to two spinless SIS′ junctions in parallel, one for each spin, and there would be a single peak in $G_{det}(V_{det})$ instead of two. The asymmetrical signal in Fig. 5 would remain in the data, but we would be unable to differentiate the contribution from the two spins and clearly identify $f_{L3}$—see Supplementary Fig. 16.) We note also that the detector current is typically 0.1–1 nA ≪ $I_{inj}$ ~ 10–100 nA throughout the subgap region: the detector is close to equilibrium.

**Non-Fermi–Dirac QP energy distributions**. Measurements at zero magnetic field already reveal non-Fermi–Dirac distributions. Figure 3a shows the current–voltage characteristics of the closest detector junction at two injection currents: 0 nA (black trace) and 120 nA (red trace). We focus on the low-voltage range (the 'subgap region') before the abrupt rise of $I_{det}$ at $V_{det} = (\Delta + \Delta_{det})/e$, where the opposite-energy coherence peaks of S and S′ align. We see that the red trace is higher than the black. This indicates the presence of additional QPs created by

injection. (Such measurements of 'excess QPs' have been made in extended junctions, but because of the spatial averaging, the spectroscopic information was lost. For a review, see ref. [22].)

This creation of QPs by current injection can also be seen in the differential conductance measurement ($G_{det}(V_{det})$) at three values of $I_{inj}$: 0, ≈13, and 120 nA (Fig. 3b). Here, we see more clearly that most of the QPs are at the gap edge ($eV_{det} = \Delta - \Delta_{det}$). If we try to fit the trace at $I_{inj} \approx 13$ nA with a thermal QP distribution, it is clear that this grossly overestimates the number of QPs at high energies (Fig. 3b, dotted line). The QPs do not thermalise.

Instead, as shown in our calculations (Fig. 1) and discussed earlier, the QP states in S are filled up to $V_{inj}$: the electron distribution function in N is 'imprinted' onto the QPs in S. This can be seen by overlaying the $I_{inj}(V_{inj})$ measurement in Fig. 3a, shifted by $\Delta_{det}/e$, on a plot of $G_{det}$ as a function of ($V_{det}$) and $I_{inj}$ (Fig. 3c). We see that, at each current, the injector voltage falls exactly at the location of a step in $G_{det}$ (seen here as a change in colour). The accumulation of QPs at the gap edge in S can also be seen on this colour scale as a yellow horizontal feature.

Our calculations reproduce both the step-like feature corresponding to $I_{inj}(V_{inj} + \Delta_{det}/e)$, as well as the horizontal feature (Fig. 3d). Thus, at a distance of about 300 nm ≪ $\lambda_{e−e}$ from the injector (i.e. at J$_{det1}$) and in the energy range of interest for the detection of the $f_{L3}$ mode, the QPs have not yet thermalised, and it is reasonable to neglect electron–electron interactions.

**Spin energy mode**. At finite magnetic fields, current injection at low energies becomes spin-polarised: we expect different distribution functions for spin up and down QPs, and in particular to excite the spin energy mode. We show in Fig. 4a calculations of $G_{det}$ as a function of $V_{det}$ (in the sub-gap region) and of $I_{inj}$, at 1 T where the density of states in S is well spin-split (Fig. 2a). Following features from low to high energies, we expect peaks in $G_{det}(V_{det})$ at $eV_{det} = (\pm|\Delta - \Delta_{det} - E_Z|)$ which we shall call $P_2$ and $P_3$, corresponding to the coherence peaks of spin down excitations (spin down electron-like or spin up hole-like QPs). Peaks at $V_{det} = \pm|\Delta - \Delta_{det} + E_Z|$ ($P_1$ and $P_4$), corresponding to the coherence peaks of spin up excitations, appear when $I_{inj}$ is increased and spin up excitations are also injected.

Comparing this to the data (Fig. 4c), we see $P_2$ and $P_3$ clearly, but $P_1$ and $P_4$ are less prominent. This is due to the increased electron–electron interaction at high energies and QP number. (For clarity, the Josephson or supercurrent contribution has been subtracted from $G_{det}$. See Supplementary Methods 1.2.2 for details.)

Next, we compare the number of electron-like and hole-like QPs by taking two slices of Fig. 4c at $eV_{det} = +|\Delta - \Delta_{det} - E_Z|$ (Fig. 4d). The traces are not identical. The difference between them, which is the charge imbalance, is maximal at $I_{inj} \approx 8$ nA, corresponding to maximal spin polarisation of the injection current, i.e. when the injection voltage is just below the coherence peak associated with spin down excitations. This charge imbalance is also reproduced in the calculation (Fig. 4b).

The charge imbalance associated with $f_{L3}$ has particular energy and magnetic field signatures: it is expected to appear in the energy range $\Delta - E_Z \leq |E| \leq \Delta + E_Z$. In Fig. 5a, we plot the component of the data in Fig. 4a which is odd in $V_{det}$, which gives the charge imbalance. The odd component is indeed largest in the expected energy range. As a function of magnetic field, the charge imbalance first becomes visible when $E_Z > 3.5k_BT$. It then continues to increase with magnetic field, as expected, then starts going down. The decrease is caused mainly by smearing of both injector and detector densities of states, due to orbital depairing. Our calculations reproduce the data at 1 T well (Fig. 5b, dash-dotted line).

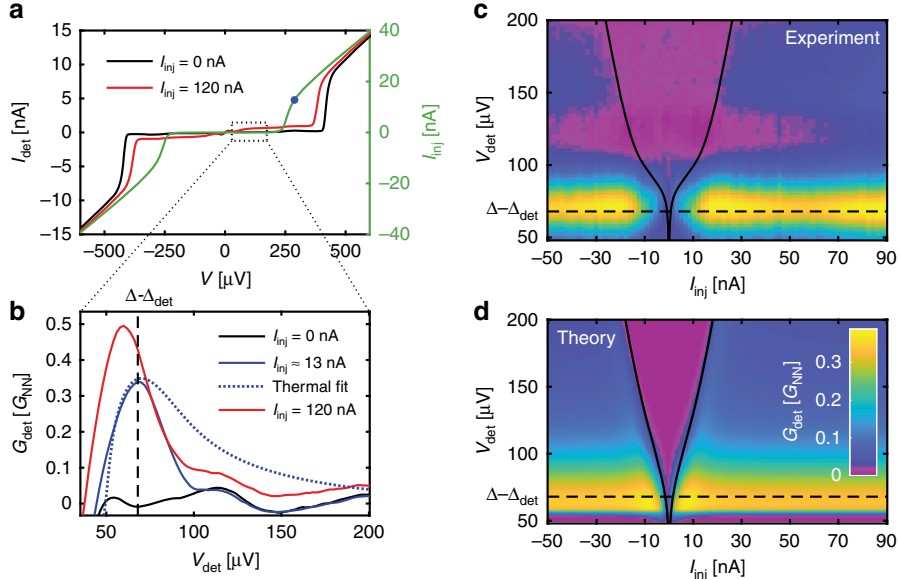

**Fig. 3 Non-Fermi–Dirac quasiparticle distribution.** The magnetic field ($H$) is zero for all panels in this figure. **a** Current ($I_{det}$) as a function of voltage ($V_{det}$) across the closest SIS' detector junction ($J_{det1}$) for injection currents $I_{inj} = 0$ nA (black) and $I_{inj} = 120$ nA (red). On the right vertical scale, $I_{inj}$ as a function of voltage $V_{inj}$ across the NIS injector junction $J_{inj}$ (green). **b** Differential conductance ($G_{det}$), in units of the normal state value, as a function of ($V_{det}$) across $J_{det1}$ for $I_{inj} = 0$ nA (black), $I_{inj} \approx 13$ nA (blue, blue dot in **a**), and $I_{inj} = 120$ nA (red). The vertical dashed line indicates $eV_{det} = \Delta - \Delta_{det}$: $G_{det}$ at this voltage is proportional to the number of quasiparticles in the superconducting wire at $E = \Delta$. An attempted fit with an effective temperature $T^* \approx 1.1$ K in S reproduces the peak at $I_{inj} = 13$ nA, but grossly overestimates the QP population at higher energies (dashed blue line). In this fit, we use the experimentally determined values $\Delta = 245$ µeV and $\Delta_{det} = 180$ µeV, $T_{det} = 90$ mK and a phenomenological depairing $\alpha \approx 0.01\Delta$. **c** $G_{det}$ at $J_{det1}$ as a function of $V_{det}$ and $I_{inj}$ with the slice at $I_{inj} = 0$ subtracted from all data. The black lines show the measurement of $\pm I_{inj}(V_{inj})$ from **a** shifted downwards by $\Delta_{det}/e$. The black lines fall at the location of a step-like feature in the colour map, as expected: as shown in Fig. 1b, QPs in S are created up $E \approx eV_{inj} + k_BT$, leading to a step-like cutoff in the distribution function. The dashed line again indicates $eV_{det} = \Delta - \Delta_{det}$, where the QP density is maximal due to the coherence peak in the DOS of S. **d** Theoretical prediction of **c**, with $\Delta$, $\Delta_{det}$ and $\alpha$ as in **b**.

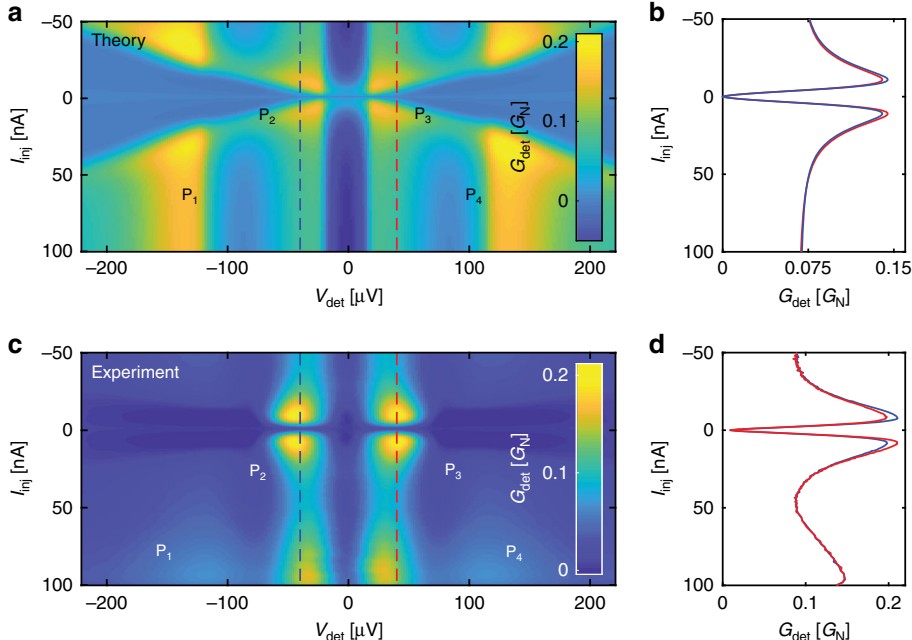

**Fig. 4 Spin energy mode.** $H = 1$ T for this figure. **a**, **c** Theoretical calculations for and measurements of the detector differential conductance as a function of detector voltage and injection current for the detector $J_{det1}$. The peaks $P_1$–$P_4$ observed experimentally and reproduced in our calculations are due to spin down ($P_2$, $P_3$) and spin up ($P_1$, $P_4$) excitations. **b**, **d** Vertical slices of **a** and **c** at $eV_{det} = \pm |\Delta - \Delta_{det} - E_Z|$ (red for + and blue for −), indicated by the dashed blue and red lines in **a** and **c**. A charge imbalance can be seen, i.e. the red and blue traces, which give, respectively, the number of electron-like and hole-like non-equilibrium quasiparticles, are not identical.

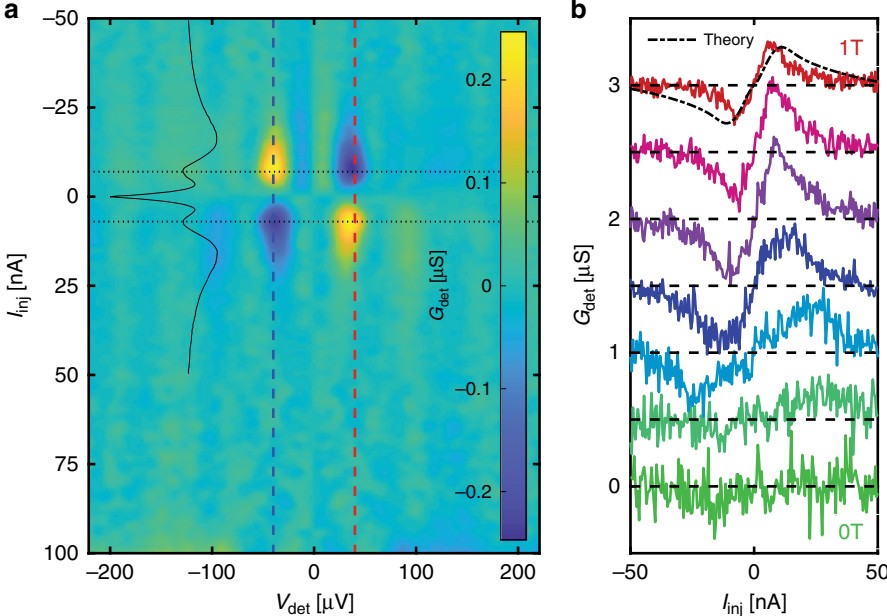

**Fig. 5 Close-up of the spin energy mode. a** The odd-in-energy (i.e. $V_{det}$) component of Fig. 4c, corresponding to a charge imbalance, which is well localised in both $V_{det}$ and $I_{inj}$. As a function of $V_{det}$, as expected, the charge imbalance only appears at the gap edge: the vertical dashed lines indicate $V_{det} = \pm |\Delta - \Delta_{det} - E_Z|/e$. As a function of the injection current, the signal is maximal (horizontal dotted lines) when only spins of one species (spin down excitations) are injected: $G_{inj}(V_{inj})$ is shown as a thin black line; its scale in the horizontal direction is in arbitrary units. **b** A vertical slice of **a** at $V_{det} = (\Delta - \Delta_{det} - E_Z)/e$ and the same measurement at different magnetic fields linearly spaced down to $H = 0$ T. The theoretical prediction for $H = 1$ T is shown in black. The charge imbalance increases slightly then decreases as the magnetic field is lowered. At $H = 0$ it is undetectable.

The odd-in-$V_{det}$ component of the data in Fig. 4b, d, which comes from $f_{L3}$, is small compared to the even-in-$V_{det}$ component, which comes from either $f_L$ or $f_{T3}$. The QPs from $f_L$ or $f_{T3}$ contribute to a finite magnetisation in the superconductor, previously detected by other methods[16,27,34–36]. At $H = 0$, we recover the previously observed charge imbalance signal[37–41], associated with the $f_T$ mode, which occurs at high energies and low magnetic fields (see Supplementary Discussion 2.5).

Beyond a spin–flip length from the injector, spin up and down QP distribution functions become identical, leading to the disappearance of $f_{L3}$. Indeed, we do not observe $f_{L3}$ at $J_{det2}$ or $J_{det3}$ (see Supplementary Discussion 2.3).

Compared to normal metals and semiconductors, the spin energy mode in superconductors has the advantage of being excitable by using the spin-split DOS. Its association with an energy-localised charge imbalance make it easy to distinguish from other modes. Using superconductors as detectors allowed us to have spectroscopic information on the QPs, by using the coherence peak in the detector density of states. This work paves the way for new spin-dependent heat transport experiments, as well as the generation of spin supercurrents by out-of-equilibrium distribution functions in conventional superconductors[18,42].

## Methods

**Sample fabrication**. The superconducting wire is 6 nm Al, while the injector is 100 nm Cu, and the detectors 8 nm Al/0.1 nm Pt. The devices were fabricated with standard electron-beam lithography and evaporation techniques. The NIS and SIS' junctions have conductances per unit area $\approx 1.9$ and $\approx 3.3$ mS $\mu m^{-2}$, respectively (corresponding to barrier transparencies of $\approx 2 \times 10^{-5}$).

**Electronic transport measurements**. All measurements were performed using standard lock-in techniques in a dilution refrigerator with a base temperature of 90 mK. The lock-in frequency is typically 17–37 Hz and the excitation voltage is 5 μV. The out-of-plane component of $H$ was compensated to be ≤1% of the total field.

## Data availability

The datasets generated during and/or analysed during the current study are available from the corresponding author upon reasonable request.

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

## Acknowledgements

We acknowledge valuable discussions with Tero Heikkilä, Mikhail Silaev and Wolfgang Belzig; and an ANR JCJC grant (SPINOES) from the French Agence Nationale de Recherche. B.Y.W. is grateful for a College of Science (CoS) Travel Grant and Scholarship from the National Taiwan University. We also thank Freek Massee, Hadar Steinberg and Suchitra Sebastian for helpful comments on the manuscript.

## Author contributions

M.K. fabricated the devices and performed the numerical calculations. M.K. and M.A. made the measurements. M.K., M.A. and C.Q.H.L. analysed the data and wrote the manuscript. B.Y.W. and M.W. were involved in earlier stages of the work.

## Competing interests

The authors declare no competing interests.
