## [Peer Review File · Nature Communications]

Reviewers' Comments:

Reviewer #1:

Remarks to the Author:

This manuscript probes spin-energy mode(s) in a superconductor via non-local transport of spin polarised quasiparticles within an un-thermalised region. The results are supported by theory involving the calculation of the distribution functions for spin up/down quasiparticles. The paper is exciting, robust and interesting to the broad readership of scientists working on superconductivity and out-of-equilibrium physics in condensed matter systems.

I am happy to support publication in Nature Communications, but request that the authors consider the following points:

1. There is a strong assumption about the quasiparticle distribution between the superconducting bar and the detector. This is potentially a weakness in the paper given the different injector-detector distances and the interfacial qualities. Could the authors comment on this in the paper.
2. Fig. 2b shows a plateau around zero energy in the absence of an injected current (no quasiparticles). What might be the origin of this plateau?
3. For Fig.5, please interpret why the charge imbalance first increases and then decreases as the field is lowered.
3. Some recent (experimental) literature on spin-transport in superconductors is missing from the references. Please check your reference list carefully.

Reviewer #2:

Remarks to the Author:

The present manuscript reports on a very interesting experiment which reveals the existence of the spin-energy mode excited in a superconductor.

Previous experiments on the non-equilibrium modes have indeed detected the other three modes (charge, spin, and energy). In this regard, the experiment complete the full picture and in a way demonstrate the validity of the theories that have been developed for the description of the non-equilibrium modes in superconductors in Zeeman fields.

Therefore I would recommend this manuscript to be published.

I have nevertheless some comments and recommendations about the the way the authors present the results. The interpretation of the experimental results strongly relies in theory and technicalities. Indeed, the authors made an effort to avoid these technicalities in the main text, however in certain parts of the main text important explanations are missing and a the reader is referred simply to the Supp. Info (SI, see below list of comments). The latter is rather long, so it would be good to know exactly where to look at. I recommend the authors to specify either section, or equations, or figures when refereeing to the SI.

The spectroscopy technique used in the present experiment to detect the spin-energy mode is in my view very interesting. And as I said above, the interpretation of the measurements strongly relies on the theoretical model which is the generalization of the Keldysh kinetic equation for superconductors in the presence of a spin-splitting field. This theory was presented in ref. [31]. As far as I know it was in that work when for the first time it was realized that due to the spin-splitting the four different modes are coupled. Indeed, the fact that there are two contributions to the charge accumulation (f_T and f_{L3}) when spin-up and spin-down DOS are different, was shown in Eq. (1) of Ref.[31]. Strangely Ref. [31] is only marginal cited in the present manuscript, when in my opinion has a central role in the interpretation of the data. The SI of the present

manuscript contains exactly the same formalism presented for the first time in the Supp. Info of Ref.[31]. It is also strange when the authors in the in the Supp. Info write:

“This chapter provides a detailed overview of the theory presented in [8] and [9], while restricting the discussion to the aspects relevant to the experiment. ”

Indeed, Ref. 8 in the Supp. Info is a review that contains among many other stuff, a detail description of the formalism introduced in ref.[31] On the other hand it is obvious that the formalism of Ref. 9 in the SI haven't be used in the present manuscript. Even though Ref. 9 introduced the concept of spin-energy mode, it did not deal with spin-splitting case and hence all kinetic coefficients were spin-independent. The spin-dependence of the kinetic coefficients (Eqs. 7-8 of the SI in the present manuscript) is crucial for understanding this and other experiments when detecting non-equilibrium modes. In my view it would be correct to emphasize the importance of the results of Ref.[31] for the present experiment.

In addition, the importance of the main finding of the present manuscript (the detection of the spin-energy mode), can be boosted by adding a brief historical account of the observation of the different modes in superconductors and the fact that the spin-energy mode still remained unrevealed.

Further comments/questions:

- 1) Why is it important that the detector is a superconductor and not a normal metal?
- 2) What would be the result in Fig. 4 if the S-detector is spin-split,?
- 3) In page 11 it is written “As expected, we do not observe f_{L3} at Jdet2 and Jdet3 ...” and a reference to the SI. is given. Maybe it is better to say with words in the main text why this is expected, or refer to the exact section of the SI where this is discussed.
- 4) Fig. 3c-d: the authors claim that “the black lines fall at the location of a step-like feature in the colour map, as expected”. I would say that this true in the theory panel d. However, in the experimental panel c there are certain regions in the (V_{det} , I_{inj}) diagram in which this does not hold. Is there a simple explanation why the experimental data is not as expected in these regions?
- 5) Minor typos: In page 10 Δ_D and Δ_{det} are used for the same quantity. The same with E_Z and $\mu_B H$
- 6) Also in page 10 it is written : “The difference between them, which is the charge imbalance, is maximal at $I_{inj} \approx 8nA$, corresponding to maximal spin polarisation of the injection current, i.e. when the injection voltage is just below the coherence peak of the second spin species. ” What does “second spin species” refer here to?

Reviewer #3:

Remarks to the Author:

The manuscript addresses an interesting question: how to experimentally resolve the spin-dependent quasiparticle populations from the different branches of the excitation spectrum of a superconductor. In previous literature, QP distributions have often been probed via superconducting tunnel spectroscopy. The present manuscript examines how a variant of this technique fares when excitations with different spin are not degenerate in energy due to a Zeeman splitting, and leverages this to observe a mixed electron-hole-spin component of the electron distribution. This tests an experimentally less explored part of the understanding of the quasiparticle aspect of superconducting spintronics. These are the novel contributions of the manuscript.

The experimental approach taken is to perform nonlocal conductance measurements in a lateral geometry, using S-probe junctions to be able to use conductance maps to resolve energy structure. To recognize the spin-dependent imbalance part, the energy and magnetic field dependency is measured, and compared to the expectations from quantum-kinetic simulations, also done by the authors here. This is a valid approach to take, and the analysis performed appears reasonable. The semi-quantitative agreement between experiment and theoretical results

is good, given the fairly simplified model.

However, I think the discussion of the two expected contributions to the charge imbalance needs to be clarified in the main text. From the discussion of Figs. 4 and 5, one gets the impression that only f_{L3} is important at large H , but this is not unambiguously stated. To clarify the situation, the authors could explicitly show both contributions (f_T and f_{L3}) to charge imbalance separately in the theory curves from their simulations, not only the total CI. The authors could perhaps also comment on the relation to the Tinkham-like excitation picture.

The manuscript is written with sufficient technical information, and the supplement clarifies details of interest to experts.

I believe the achievement is interesting enough for publication in Nat. Comms., and that a suitably revised manuscript could be considered for publication.

Technical comments:

- In Fig. 1(a-b), what quantity are the shaded curves more precisely? Excess quasiparticles? The x-axis indicates $N_{up/down}$, but they probably are not $N = \rho * f$ from panels (a) and (c) as in Eq. (1). It's also somewhat unclear what the "multiplied five times" in the caption means. I think the illustration needs to be revised.
- The assumption is that the detector electrodes are not spin-split by external field, due to the Pt layer. How sensitive are the results vs. this assumption, e.g., based on allowing for small spin splitting in their theory model?
- In Eq. (3), $f(E)$ should probably be $f_{noneq}(E)$ as in the supplement? Also check the sign of V in (3).
- For accessibility to non-expert readers, the qualitative description of the spectroscopy method should explain also the $V_{det} < 0$ side, so that the discussion of the symmetry of the figures can be understood in comparison to that in Eq. (1).
- On p. 11 "odd component of the data" --- odd in which quantity? "... which comes from f_{L3} " --- at this point in the discussion, this point has not yet been established.
- Fig. 5a) caption has reference to "top axis", but there is no scale or labels on the top. I'd recommend the authors to carefully check the main text in entirety.
- In Ref. [41], "such measurements of excess QPs have been made in extended junctions" --- which references?

Responses to Reviewer #1:

This manuscript probes spin-energy mode(s) in a superconductor via non-local transport of spin polarised quasiparticles within an un-thermalised region. The results are supported by theory involving the calculation of the distribution functions for spin up/down quasiparticles. The paper is exciting, robust and interesting to the broad readership of scientists working on superconductivity and out-of-equilibrium physics in condensed matter systems.

I am happy to support publication in Nature Communications, but request that the authors consider the following points:

1. There is a strong assumption about the quasiparticle distribution between the superconducting bar and the detector. This is potentially a weakness in the paper given the different injector-detector distances and the interfacial qualities. Could the authors comment on this in the paper.

The spatial evolution of the QP distribution function in the superconducting wire was explored experimentally (Supplementary Sections 3.1 and 3.3). We observe a linear decay of the number of quasiparticles from the injection point, which agrees with the theory of Supplementary Section 1. This indicates negligible leakage of quasiparticles from or to the detector.

Regarding the boundary condition at the detector (which forms an SIS' tunnel junction with the superconducting wire): quasiparticles are much more likely to diffuse along the superconducting wire than to leak into the detector. In addition, throughout the subgap region, the detector current is typically 0.1 – 1 nA. This is much smaller than the injector current of 10 – 100nA. Thus, it is reasonable to neglect any contribution from the detector to the out-of-equilibrium QP population in the superconducting wire, and also to assume that the detector is at equilibrium. This is noted on p. 8 of the manuscript.

2. Fig. 2b shows a plateau around zero energy in the absence of an injected current (no quasiparticles). What might be the origin of this plateau?

We believe the referee is referring to the peak at zero detector voltage, which is due to the Josephson effect. In this study, it is the quasiparticles rather than the Cooper pairs which are of interest. Thus, we do not focus on the Josephson signal, which is nevertheless discussed in some detail in Supplementary Section 2.2.1.

3. For Fig.5, please interpret why the charge imbalance first increases and then decreases as the field is lowered.

The charge imbalance associated with the spin energy mode (f_{L3}) first increases with magnetic field as the Zeeman energy, which is linear in field, increases. The increase of the Zeeman energy increases the maximum number of spin-polarised quasiparticles that we can inject. The energy range over which f_{L3} is visible also increases. As can be seen in Figure 5, it first becomes detectable when the Zeeman energy exceeds the temperature. At higher magnetic fields, the signal decreases due mainly to smearing of both injector and detector densities of states, because of orbital depairing. We have incorporated this into the main text.

4. Some recent (experimental) literature on spin-transport in superconductors is missing from the references. Please check your reference list carefully.

We have reviewed the recent literature on spin-dependent thermal and thermoelectric effects in superconductors and updated our references. As spin transport in superconductors is quite a broad

topic, we may have missed recent publications relevant to our work. If the reviewer is aware of any such work, we would be happy if s/he could let us know about it.

Responses to Reviewer #2:

The present manuscript reports on a very interesting experiment which reveals the existence of the spin-energy mode excited in a superconductor.

Previous experiments on the non-equilibrium modes have indeed detected the other three modes (charge, spin, and energy). In this regard, the experiment completes the full picture and in a way demonstrates the validity of the theories that have been developed for the description of the non-equilibrium modes in superconductors in Zeeman fields.

Therefore I would recommend this manuscript to be published.

I have nevertheless some comments and recommendations about the way the authors present the results. The interpretation of the experimental results strongly relies on theory and technicalities. Indeed, the authors made an effort to avoid these technicalities in the main text, however in certain parts of the main text important explanations are missing and the reader is referred simply to the Supp. Info (SI, see below list of comments). The latter is rather long, so it would be good to know exactly where to look at. I recommend the authors to specify either section, or equations, or figures when referring to the SI.

We thank the referee for this helpful suggestion, which we have implemented.

The spectroscopy technique used in the present experiment to detect the spin-energy mode is in my view very interesting. And as I said above, the interpretation of the measurements strongly relies on the theoretical model which is the generalization of the Keldysh kinetic equation for superconductors in the presence of a spin-splitting field. This theory was presented in ref. [31]. As far as I know it was in that work when for the first time it was realized that due to the spin-splitting the four different modes are coupled. Indeed, the fact that there are two contributions to the charge accumulation ($f_{\uparrow T}$ and $f_{\downarrow L}$) when spin-up and spin-down DOS are different, was shown in Eq. (1) of Ref.[31]. Strangely Ref. [31] is only marginally cited in the present manuscript, when in my opinion has a central role in the interpretation of the data. The SI of the present manuscript contains exactly the same formalism presented for the first time in the Supp. Info of Ref.[31]. It is also strange when the authors in the in the Supp. Info write: "This chapter provides a detailed overview of the theory presented in [8] and [9], while restricting the discussion to the aspects relevant to the experiment." Indeed, Ref. 8 in the Supp. Info is a review that contains among many other things, a detailed description of the formalism introduced in ref.[31] On the other hand it is obvious that the formalism of Ref. 9 in the SI haven't been used in the present manuscript. Even though Ref. 9 introduced the concept of spin-energy mode, it did not deal with spin-splitting case and hence all kinetic coefficients were spin-independent. The spin-dependence of the kinetic coefficients (Eqs. 7-8 of the SI in the present manuscript) is crucial for understanding this and other experiments when detecting non-equilibrium modes. In my view it would be correct to emphasize the importance of the results of Ref.[31] for the present experiment.

The referee is right that the 2015 PRL paper by Silaev et. al. first derived the equations for spin-dependent transport in Zeeman-split superconductors. To develop our understanding of the theory we relied heavily on the review article by Heikkilä et al. (2019); the original order of citations resulted from this learning process, which we then tried to convey to the reader, losing sight of scientific priority in the process. We have now modified both the main text and the Supp. Info. to emphasize that the ideas first proposed by Silaev et al. are crucial to our work.

In addition, the importance of the main finding of the present manuscript (the detection of the spin-energy mode), can be boosted by adding a brief historical account of the observation of the different modes in superconductors and the fact that the spin-energy mode still remained unrevealed.

We attempted a historical account at the beginning of the section entitled ‘Spinful Excitation Modes of Out-of-Equilibrium Superconductors’. We have now added a sentence with references to previous work on charge and energy modes. Note that the spin mode has still not been unambiguously identified experimentally: while it is clear that previously observed spin imbalances within a spin-flip length of the injector must have contributions from the spin mode, the f_L and f_{T3} contributions have not been clearly separated as been done for f_T and f_{L3} in the present manuscript (whether based on energy/magnetic field dependences or by some other method). In this sense, while this would certainly ‘add value’ to the presentation of our results, we do not think it completely accurate to say that ‘only f_{L3} remained unrevealed’, if indeed that is what is suggested. We nevertheless thank the referee for thinking about how to boost the visibility of our results.

1) Why is it important that the detector is a superconductor and not a normal metal?

A superconducting detector is a natural energy filter: the peak in its density of states ‘picks out’ the number of quasiparticles at a given energy and allows us to perform spectroscopy of the quasiparticle population. This spectroscopic information would be lost in the case of a normal detector.

2) What would be the result in Fig. 4 if the S-detector is spin-split,?

Were there a Zeeman splitting in S’ equal to that in S, the situation would be equivalent to two spinless SIS junctions in parallel, one for each spin. Thus, the peaks P1 and P2 in Figure 4 would be merged as would be P3 and P4. In this case, we would still measure the charge imbalance due to f_{L3} but would not be able to tell which spin species it is associated with. Indeed, this is what was measured in samples of a previous generation, before we moved to the non-spin-split superconducting detectors used to obtain the data shown in the manuscript (cf. Supplementary Figure 16).

3) In page 11 it is written “As expected, we do not observe f_{L3} at Jdet2 and Jdet3 ...” and a reference to the SI. is given. Maybe it is better to say with words in the main text why this is expected, or refer to the exact section of the SI where this is discussed.

We have expanded this explanation and now also refer to Supplementary Section 3.3, which contains additional data.

4) Fig. 3c-d: the authors claim that “the black lines fall at the location of a step-like feature in the colour map, as expected”. I would say that this true in the theory panel d. However, in the experimental panel c there are certain regions in the (V_{det}, I_{inj}) diagram in which this does not hold. Is there a simple explanation why the experimental data is not as expected in these regions?

Panel c of figure 3 in the main text indeed shows a depressed (purple) region outside of the bounds indicated by $I_{inj}(V_{inj})$. To highlight the step-like feature of the nonthermal distribution we subtracted the equilibrium by $G_{det}(V_{det})$ trace (black line, Figure 3b) from the data. The purple region is at $V_{det} \approx 120\mu\text{V}$, which is where there is a peak in the background trace, due to the Josephson contribution to the detector current. As the system is driven out of equilibrium the Josephson coupling is reduced, which also reduces the height of this peak. Thus when the subtraction is performed there is an ‘erroneously’ depressed region.

The same features can also be observed at higher fields (dark blue region of Figure 4c). To highlight this a new section (3.2) has been introduced in the Supp. Info. which shows this directly, along with an explanation of the artefact in Figure 2 of the main text.

5) *Minor typos: In page 10 Delta_D and Delta_det are used for the same quantity. The same with E_Z and mu_B H*

We thank the referee for pointing out these typos, which have been fixed.

6) *Also in page 10 it is written : “The difference between them, which is the charge imbalance, is maximal at $I_{inj} \approx 8nA$, corresponding to maximal spin polarisation of the injection current, i.e. when the injection voltage is just below the coherence peak of the second spin species. “What does “second spin species” refer here to?*

We have changed this to read ‘the coherence peak associated with spin down excitations’, which we hope is clearer. (Spin up excitations occur at lower energy than spin down ones.)

Responses to Reviewer #3:

The manuscript addresses an interesting question: how to experimentally resolve the spin-dependent quasiparticle populations from the different branches of the excitation spectrum of a superconductor. In previous literature, QP distributions have often been probed via superconducting tunnel spectroscopy. The present manuscript examines how a variant of this technique fares when excitations with different spin are not degenerate in energy due to a Zeeman splitting, and leverages this to observe a mixed electron-hole-spin component of the electron distribution. This tests an experimentally less explored part of the understanding of the quasiparticle aspect of superconducting spintronics. These are the novel contributions of the manuscript.

The experimental approach taken is to perform nonlocal conductance measurements in a lateral geometry, using S-probe junctions to be able to use conductance maps to resolve energy structure. To recognize the spin-dependent imbalance part, the energy and magnetic field dependency is measured, and compared to the expectations from quantum-kinetic simulations, also done by the authors here. This is a valid approach to take, and the analysis performed appears reasonable. The semi-quantitative agreement between experiment and theoretical results is good, given the fairly simplified model.

However, I think the discussion of the two expected contributions to the charge imbalance needs to be clarified in the main text. From the discussion of Figs. 4 and 5, one gets the impression that only f_{L3} is important at large H, but this is not unambiguously stated. To clarify the situation, the authors could explicitly show both contributions (f_T and f_{L3}) to charge imbalance separately in the theory curves from their simulations, not only the total CI.

To respond to this suggestion, we refer the referee to Supp. Sections 1.3.3, 1.4, 1.5 and 3.5.

Equation 20 (Section 1.3.3) expresses the different nonequilibrium components of the distribution function in terms of four χ coefficients. The bottom panel of Figure 5 shows χ_T and χ_{L3} , which are related to the charge imbalance. Indeed, close to the gap edge the charge imbalance is an (almost) equal mixture of the two modes. This is a result of the coupled transport and relaxation of the two modes. Supp. Section 1.4, in particular Equation 22, shows that the total charge

imbalance, from which one can see that the charge imbalance associated with the spin-energy mode is well localized close to the gap edge.

In Supp. Figure 6 (Supp. Section 1.5), we show the theoretical contributions of f_T and f_{L3} to the charge mode, as a function of injection voltage and magnetic field. Here it can be seen that the f_{L3} contribution is limited to a specific energy range and grows with magnetic field, whereas the f_T contribution is more important at high energies and low magnetic fields.

In Supp. Section 3.5 we show additional data: At zero magnetic field and at energies beyond the gap edge, a charge imbalance can be seen. This charge imbalance disappears at a higher magnetic field (1 T). This is all as expected for the f_T mode.

The authors could perhaps also comment on the relation to the Tinkham-like excitation picture.

The general situation we consider is illustrated by Figure 1, in which both charge and energy imbalances are present – arising from f_T and f_{L3} for the charge imbalance, and f_L and f_{T3} for the energy imbalance. The Tinkham excitation picture (presented e.g. in Chapter 11 of *Introduction to Superconductivity*) applies to non-equilibrium superconductivity at zero magnetic field. He considered quasiparticles to be spinless and therefore charge and energy imbalances arose only from (respectively) f_T and f_L – the ‘odd’ and ‘even’ modes in his nomenclature.

We have tried to make it clearer in the main text and the Supp. Info. that the spinful excitations f_{L3} and f_{T3} were introduced in Morten et al. PRB (2004) and the case of finite Zeeman field was first explored by Silaev et al. PRL (2015). In the latter, one can find a pictorial representation similar to the one in Tinkham’s book.

Instead of using this illustration, we have chosen to introduce the spin energy mode via Figure 1, which shows that (a) spin up and down distribution functions are different, (b) we expect a peak in the number of quasiparticles at the gap edge, (c) there is a step-like cut-off in the number of quasiparticles at V_{inj} , and (d) there is a charge imbalance associated with f_{L3} . We feel that this would be more helpful for explaining our experimental results.

The manuscript is written with sufficient technical information, and the supplement clarifies details of interest to experts.

I believe the achievement is interesting enough for publication in Nat. Comms., and that a suitably revised manuscript could be considered for publication.

*- In Fig. 1(a-b), what quantity are the shaded curves more precisely? Excess quasiparticles? The x-axis indicates $N_{up/down}$, but they probably are not $N = \rho * f$ from panels (a) and (c) as in Eq. (1).*

The referee is right. The shaded curves are in fact $N_{\uparrow}(E) = \rho_{\uparrow}(E)[f_{\uparrow}(E) - f_{\uparrow}^0(E)]$ and $N_{\downarrow}(E) = \rho_{\downarrow}(E)[f_{\downarrow}(E) - f_{\downarrow}^0(E)]$ calculated from panels (a) and (c). $f_{\uparrow}^0(E)$ and $f_{\downarrow}^0(E)$ are respectively $f_{\uparrow}(E)$ and $f_{\downarrow}(E)$ at equilibrium. To address this and other issues noted below, we have revised Equations 1 to 3 and the associated text.

It's also somewhat unclear what the "multiplied five times" in the caption means. I think the illustration needs to be revised.

The charge imbalance, i.e. the odd component of $N_{\uparrow}(E) + N_{\downarrow}(E)$ has been multiplied five times to that it can be more clearly seen that there are more blue quasiparticles than red ones. We have revised the caption and hope that it is now clearer.

- The assumption is that the detector electrodes are not spin-split by external field, due to the Pt layer. How sensitive are the results vs. this assumption, e.g., based on allowing for small spin splitting in their theory model?

That the detector electrodes are not spin-split by the applied field is not an assumption but is based on measurements. (See Supplementary Section 2.3.) The unsplit coherence peak in the detector is due to spin mixing by the sprinkled Pt. It allows us to ‘pick out’ the number of quasiparticles at a particular energy, and thus perform spin-resolved spectroscopy, as quasiparticles with different spins in S appear at different energy. Were the detector not spin-split, the situation would be equivalent to two spinless SIS junctions in parallel, one for each spin. Thus, the peaks P1 and P2 in Figure 4 would be merged as would be P3 and P4. In this case, we would still measure the charge imbalance due to f_{L3} but would not be able to tell which spin species it is associated with. Indeed, this is what was measured in samples of a previous generation, before we moved to the non-spin-split superconducting detectors used to obtain the data shown in the manuscript (cf. Supplementary Figure 16).

- In Eq. (3), $f(E)$ should probably be $f_{\text{noneq}}(E)$ as in the supplement? Also check the sign of V in (3).

The referee is right. We have revised Equations 1 to 3 and the associated text to address this issue.

- For accessibility to non-expert readers, the qualitative description of the spectroscopy method should explain also the $V_{\text{det}} < 0$ side, so that the discussion of the symmetry of the figures can be understood in comparison to that in Eq. (1).

We have revised Equations 1 to 3 and the associated text to clarify the connection between the spin energy (f_{L3}) mode, charge imbalance and the symmetry of $G(V_{\text{det}})$. The main change in Equations 1 and 2 is the subtraction of equilibrium particles, so that we focus only on the excitations. This allows us make the connection to $G(V_{\text{det}})$ in Equation 3 clearer, and to highlight its symmetry properties. In addition, we have expanded Supplementary Section 2.2.2 to address this issue.

- On p. 11 "odd component of the data" --- odd in which quantity? "... which comes from f_{L3} " --- at this point in the discussion, this point has not yet been established.

We have clarified that we are referring to the odd-in- V_{det} component of the data. In addition, we hope that the new discussion around Equations 1 to three help to clarify this point before p. 11.

- Fig. 5a) caption has reference to "top axis", but there is no scale or labels on the top. I'd recommend the authors to carefully check the main text in entirety.

The referee is right: the horizontal scale is in arbitrary units. We have corrected the caption to reflect this.

- In Ref. [41], "such measurements of excess QPs have been made in extended junctions" --- which references?

We have added a reference to a review article which provides a good overview of relevant experiments in extended junctions.

Reviewers' Comments:

Reviewer #1:

Remarks to the Author:

Many thanks to the authors for carefully and diligently responding to my comments and the comments made by the other referees. I am happy with the paper, rebuttal and supplementary materials. This is an outstanding result in the field of superconducting spintronics and I am delighted to support publication in Nature Communications.

Reviewer #2:

Remarks to the Author:

The authors addressed all my comments satisfactory and I recommend the manuscript for publication.

Reviewer #3:

Remarks to the Author:

The authors have addressed the points I raised in my initial review in their reply, and clarified relevant issues in the revised manuscript text.

The authors are correct that the relevant information about the f_T/f_{L3} modes are explained in the Supplement, for expert readers interested in this question. The current manuscript text also now more clearly refers readers to its appropriate sections also on this topic.

I can recommend the present version of the manuscript to be published in Nat. Comms.